# Studies on the Interaction of Alyteserin 1c Peptide and Its Cationic Analogue with Model Membranes Imitating Mammalian and Bacterial Membranes

**DOI:** 10.3390/biom9100527

**Published:** 2019-09-25

**Authors:** Alberto Aragón-Muriel, Alessio Ausili, Kevin Sánchez, Oscar E. Rojas A., Juan Londoño Mosquera, Dorian Polo-Cerón, Jose Oñate-Garzón

**Affiliations:** 1Facultad de Ciencias Naturales y Exactas, Departmento de Química, Laboratorio of Investigación en Catalisis and Procesos (LICAP), Universidad del Valle, Cali 760001, Colombia; alberto.aragon@correounivalle.edu.co (A.A.-M.); david.londono@correounivalle.edu.co (J.L.M.); dorian.polo@correounivalle.edu.co (D.P.-C.); 2Departmento de Bioquímica y Biología Molecular-A, Facultad de Medicina Veterinaria, Campus of International Excellence Mare, Universidad de Murcia, E-30100 Murcia, Spain; aausili@um.es; 3Grupo de Investigación en Química y Biotecnología (QUIBIO), Facultad de Ciencias Básicas, Universidad Santiago de Cali, Cali 760031, Colombia; kevinsan113@hotmail.com (K.S.); oerojas@usc.edu.co (O.E.R.A.)

**Keywords:** alyteserin 1c, model membranes, calorimetry, infrared spectroscopy, molecular dynamics

## Abstract

Antimicrobial peptides (AMPs) are effector molecules of the innate immune system and have been isolated from multiple organisms. Their antimicrobial properties are due to the fact that they interact mainly with the anionic membrane of the microorganisms, permeabilizing it and releasing the cytoplasmic content. Alyteserin 1c (+2), an AMP isolated from *Alytes obstetricans* and its more cationic and hydrophilic analogue (+5) were synthesized using the solid phase method, in order to study the interaction with model membranes by calorimetric and spectroscopic assays. Differential scanning calorimetry (DSC) showed that both peptides had a strong effect when the membrane contained phosphatidylcholine (PC) alone or was mixed with phosphatidylglycerol (PG), increasing membrane fluidization. Attenuated total reflectance Fourier transform infrared spectroscopy (ATR-FTIR) was used to study the secondary structure of the peptide. Peptide +2 exhibited a transition from β-sheet/turns to β-sheet/α-helix structures after binding with model membranes, whereas peptide +5 had a transition from aggregation/unordered to β-sheet/α-helix structures after binding with membrane-contained PC. Interestingly, the latter showed a β-sheet structure predominantly in the presence of PG lipids. Additionally, molecular dynamics (MD) results showed that the carboxy-terminal of the peptide +5 has the ability to insert into the surface of the PC/PG membranes, resulting in the increase of the membrane fluidity.

## 1. Introduction

AMPs are defined as effector molecules of the innate immune system becoming isolated from multiple organisms and have a broad spectrum of activity against bacteria, viruses, fungi, parasites, and even cancer cells [1]. Their antimicrobial targets include anionic membranes of microbes affecting them from an alteration of packing of the phospholipids to membrane disaggregation. Consequently, membrane integrity is lost, and cytoplasmic content is released to the outside by diffusion, dissipating potential membrane and killing the microbes [2]. Microorganisms can barely survive to such effects due to the metabolic “cost” required for membrane repair [3]. Thus, AMPs can be postulated as alternative therapeutic agents to mitigate the problem of microbial resistance to conventional antibiotics.

AMPs have the propensity to form an amphipathic α-helix in the environment of a phospholipid vesicle or in a membrane-mimetic solvent such as 50% trifluoroethanol–water [4]. In addition to helix conformation, AMPs can also acquire the form of β-sheets, turns, and random coils depending on the bulk solvent environment [5]. Structural properties such as amphipathicity, hydrophobicity, helicity, and charge are intimately related with antimicrobial activity [2,6]. An amphipathic α-helix is one of the most efficient structures with hydrophobic residues down one side of the helix and cationic/hydrophilic residues down the other side, both sides being important for antibacterial activity [7,8]. Net charge at the polar face is another essential structural property for the antimicrobial effect since basic residues at the peptide sequence are responsible for the initial electrostatic attraction between peptides and anionic microbial membranes [9]. In addition, there is evidence suggesting that anionic lipopolysaccharide (LPS) and lipotheicoic acid found in the outer envelope of Gram-negative and of Gram-positive bacteria, respectively, contribute in such initial attraction [10]. After the threshold concentration of the peptide is reached on the membrane surface, the hydrophobic residues of the nonpolar face are inserted into the hydrophobic core membrane [11].

The 23 amino acid residue peptide with a net charge of +2, Alyteserin-1c (+2), was first isolated from norepinephrine-stimulated skin secretions from the midwife toad *Alytes obstetricans*. Inhibitory activity against Gram-negative bacteria and low hemolytic activity against human erythrocytes (LC50 = 220 μM) were reported for peptide +2 [12]. Its cationic analogue, peptide +5, has also shown antibacterial activity [13]. The structure of peptide +2 was investigated in solution and in membrane, mimicking environments by proton NMR spectroscopy, suggesting a structure characterized by an extended alpha helix between residues LEU2 and VAL21 at a similar environment to membranes [14]. 

However, the effect of AMP on bacterial membranes is still poorly understood. Synthetic model membranes have been widely used for studying the peptide–membrane interaction and the action mechanisms of AMP [15,16,17,18,19] since those simplify the complexity of cell membranes omitting the contributions of glycoproteins and other membrane components on the peptide activity. Furthermore, different variables such as pH, ionic strength, and composition can be controlled by using model membranes. Biophysical studies on model membranes can give valuable insights into the mechanism of action of the AMPs and there are several reports about the lipid–peptide interaction using techniques like calorimetry and spectroscopy [18,20,21], while molecular dynamics (MD) simulations can provide a complementary approach to study the action mechanism in detail [22]. 

In this research, a peptide +5 from Alyteserin 1c (2+) was also designed, where hydrophobic amino acids were replaced by hydrophilic amino acids at the polar face of the helix, increasing the amphipathicity and decreasing the hydrophobicity. In addition, a substitution of an anionic amino acid by a cationic one was performed in order to increase the charge and hydrophilicity at the polar face. Thermodynamic profiles of the phase transition of model membranes imitating bacteria and erythrocytes surfaces, were studied under peptides effects using differential scanning calorimetry (DSC). Additionally, characterization of the peptide–lipid interactions was performed by attenuated total reflectance Fourier transform infrared spectroscopy (ATR-FTIR). Finally, we used a series of MD simulations over microsecond timescales on Alyteserin-1c and its analogue to systematically investigate the interaction with accurate lipid bilayer models, which mimics the surface charge of Gram-negative bacterial membranes.

## 2. Materials and Methods

### 2.1. Materials

Lipids, 1-palmitoyl-2-oleoyl-sn-glycero-3-phosphoglycerol (POPG), 1,2-dimyristoyl-sn-glycero-3-phosphoglycerol (DMPG), 1-palmitoyl-2-oleoyl-sn-glycero-3-phosphocholine (POPC), and 1,2dimyristoyl-sn-glycero-3-phosphocholine (DMPC), were purchased from Sigma (St. Louis, MO, USA) and used without further purification.

### 2.2. Peptide Synthesis and Purification

Alyteserin 1c (GLKEIFKAGLGSLVKGIAAHVAS; +2) and its analogue (GLKRIFKSGLGKLVKGISAHVAS; +5) were synthesized using a standard Fmoc peptide synthesis protocol [13]. After peptide elaboration, purification was achieved by applying reverse-phase high-resolution liquid chromatography using a semi-preparative Chromolith^®^ RP-18e column and applying a mixture of (A) H_2_O with 0.05% TFA (*v*/*v*) and (B) acetonitrile containing 0.05% TFA (*v*/*v*) as the mobile phase. For the elution of peptides, the following programed gradient was used: 70 min with 0%–50% B at 3 mL·min^−1^ and detection at 220 nm. The samples were dried using a RapidVap^®^ (Labconco, Missouri, USA) and then lyophilized to remove any remaining solvent. Finally, the molar mass of purified peptides was determined by matrix-assisted laser desorption ionization-time of flight MS using a Bruker Daltonics mass spectrometer (Bruker Daltonics Inc. Billerica, USA).

### 2.3. Helical Wheel Projection of the Peptides

Helical wheel projections of both peptides were performed using the web server HeliQuest (http://heliquest.ipmc.cnrs.fr/) and built with 18 residues of the NH-terminal sequence, since it was the region where amino acid substitutions were performed. 

### 2.4. Model Membranes Preparation

Dehydrated anionic phosphatidylglycerol (PG) and zwitterionic phosphatidylcholine (PC) lipids were dissolved in chloroform/methanol (2:1 *v*/*v*). Then, sufficient amounts of PC and PC/PG (3:1 or 1:1) mixtures which mimic the mammalian and bacterial membranes, respectively [23], were dried under a gentle current of nitrogen and placed under vacuum for 3 h to remove any residual solvent. The lipid films were hydrated with HEPES buffer (25 mM HEPES, pH 7.0, 100 mM NaCl, 0.2 mM EDTA), vigorously vortexed for 2 min, and incubated for 10 min at 37 °C above the temperature of phase transition (Tm) by three times obtaining multilamellar vesicles (MLVs) [19].

### 2.5. Differential Scanning Calorimetry (DSC)

DSC analysis was done using a TA instrument (DSC Q25). MLVs were prepared using 2 mg of lipid and three peptide dilutions to give different peptide–lipid ratios: 1:100, 1:50, and 1:25; HEPES buffer was used as a reference solution. Samples were encapsulated in standard aluminum DSC pans and the scanning was carried out over an 8–40 °C temperature at a heating rate of 1 °C min^−1^. Thermograms were acquired and analyzed using Trios software package (TA Instruments) in order to obtain the temperature of phase transition (Tm) and the transition enthalpy (ΔH). 

### 2.6. Attenuated Total Reflectance-Fourier Transform Infrared (ATR-FTIR)

POPC and POPC/POPG (1:1) MLVs were prepared using 1 mg of lipid with peptide to achieve a peptide–lipid ratio of 1:10. Unsaturated lipids were used to study the effect of the peptides on model membranes in fluid phase. MLVs were generated by hydrating the lipids with the same buffer as described above but using deuterated water. An appropriate amount of lyophilized peptide was added to the liposomes and mixed. Samples were directly placed on a Shimadzu Affinity 1 spectrophotometer equipped with a single reflection diamond ATR accessory, and they were partially air-dried at room temperature for 10 min. The spectra were collected maintaining the samples sufficiently hydrated in a saturated deuterium water vapor atmosphere. A spectrum of air was subtracted from all the sample spectra and a 10 points Adjacent-Averaging smoothing was applied to the resulting spectra in order to reduce the noise due to the aqueous vapor. The spectra were normalized between 1700 and 1600 cm^−1^ to the same area value and a baseline of 1800–1300 cm^−1^ was subtracted from all the spectra. Second derivative spectra were calculated over a 15 data-point range (15 cm^−1^).

The quantitative analysis of the peptide’s secondary structure was performed by curve-fitting of the amide I’ band [24] using the peak fitting module of the Origin software (OriginLab Corporation, Northampton, MA, USA). The band shape was set to a Voigt profile, and the fitting was obtained by iteration in two steps, the first iteration was performed fixing the peak positions as obtained by second derivative spectra, while in the second step, the bands were free [25].

### 2.7. Molecular Modelling and Molecular Dynamics Simulation of the Complex Peptide-Membrane

The molecular design of the peptide +5 was performed using a homology-based modelling procedure and the amino acid sequence from Alyteserin-1c (peptide +2, PDB code 2L5R), used as a template for the amino acid substitutions. The side-chain modeling was carried out using the DeepView Swiss-PdbViewer software (http://www.expasy.org/spdbv/) [13,26]. On the other hand, the initial configuration for the molecular dynamic simulation of the lipid bilayer conformation was modeled in a double layered spherical vesicle POPC and POPG with water inside, using the Packmol software (http://m3g.iqm.unicamp.br/packmol/home.shtml) [27]. At the modeled vesicle, 2.0 angstroms (Å) was used as a tolerance measure, meaning that all the atoms surrounding POPG were at least 2 Å away from existing atoms. The system created was constituted by a POPC/POPG mixture at a molar ratio of 3:1, besides a small sphere of water molecules inside the vesicle. The lipid bilayer was built with a set of 924 water molecules oriented in a middle sphere, centered in the origin; a POPC set containing 270 and 900 molecules for the inner and outer monolayer, respectively; and a POPG set containing 90 and 300 molecules for inner and outer monolayer, respectively. The total number of molecules in the vesicle was 1560.

The complex vesicle-peptide was performed placing the peptide in a horizontal position above the external layer. Using the VMD software (https://www.ks.uiuc.edu/Research/vmd/) the residues in the peptide were manually configured as less than 3.0 Å with respect to POPC and POPG phospholipids. Each individual model was submitted to energy minimization, and the dihedral angles, bonds, and spatial conformation were not modified in a new complex created. The water solvation was developed using TIP3 implicit water model simulation [28] with a 10 Å rigid cubic cell measured from the atom with the largest coordinate in every direction, (Vector 1: 196.7 Å, Vector 2: 199.3 Å, and Vector 3: 194.1 Å). 

Finally, three models were designed for the molecular dynamics simulation. The first model consisted of a free vesicle (POPC and POPG phospholipids) without peptide, and every other model consisted of a complex of vesicle with each peptide. Each model was solvated in a water box conformation and configured by the empirical force field parametrization that is coordinated with the program CHARMM (Chemistry at HARvard Macromolecular Mechanics): CHARMM22 all-hydrogen parameter file for proteins and lipids [29,30] and CHARMM36 all-hydrogen parameter file for proteins and lipids [31,32]. The energy of all the complex models was minimized every 100 steps with a 12 Å cutoff. The MD simulation visualization file was set to 250 steps per visualization. The simulation time was 500 ps, each step for 1 fs (500,000 steps). Temperature and pressure were 37 °C and 1 atm, respectively, achieving an isobaric-isothermal ensemble. Visual Molecular Dynamics (VMD1.9.3, for visualization) and the Scalable Molecular Dynamic (NAMD2.12) simulation software was used [33].

The phase point equilibrium runs the simulation until properties get stable with respect to time. Average energy routine parameters were calculated as stated in Equation (1):(1)$$E=\frac{1}{N}\overset{N}{\underset{i=1}{\sum }}Ei$$
and *RMSD (Root Mean Square Deviation)* as in Equation (2):(2)$$RMSD=\sqrt{\overset{N\;atoms}{\underset{i=1}{\sum }}\frac{{\left(ri\left(t1\right)-ri\left(t2\right)\right)}^{2}}{N\;atoms}}$$
where *N atoms* is the numbers of atoms whose positions are being compared, and *ri*(*t*) is the position of atom *i* at the time *t*.

## 3. Results and Discussion

### 3.1. Peptide Sequence

The peptide template sequence used for the development of this work was the peptide Alyteserin 1c (H0USY4, code UniProt KB), where only the polar side of the helix was modified (Figure 1), substituting 2 alanines for two serines, one glutamic acid for arginine, and one serine for lysine. These modifications were made according to the suggestions by Bordo and Argos [34] in order to alter the structural and physicochemical properties of the peptide as little as possible. However, substitutions of hydrophobic amino acids by polar amino acids, inevitably increases the hydrophilicity and, in turn, the amphipathicity of the peptide, while the hydrophobicity is reduced [13].

### 3.2. Thermotropic Behavior of Model Membranes

The peaks of the main transition from gel (lβ’) to crystalline liquid (lα) phase exhibited an endothermic behavior (Figure 2). The pre-transition and main transition temperature (Tm) of the MLVs constituted by DMPC was 12.9 and 22.9 °C, respectively (Table 1). Both peptides considerably affected the peak of the main transition, and the pre-transition of MLVs contained DMPC. The latter peak disappeared at the lowest concentration of peptide (1:100 peptide–lipid molar ratio), suggesting a disturbance on these MLVs with a minimum amount of peptide, since the pre-transition is very sensitive to strange molecules [35]. By increasing the amount of peptide, the Tm was reduced from 22.9 to 21.2 and 20.5 °C (Table 1) at the maximum concentration of peptides +2 and +5, respectively. 

Both peptides had the ability to decrease the order of the bilayer and to increase the fluidity of the membrane after the peptide–lipid interaction. Furthermore, a rise of domains within the membrane with lower transition temperatures was observed while the peptide concentration was increased (Figure 2 and Figure 3), suggesting that there was separation of the membrane into domains rich in peptide (the shoulder at a lower temperature) and a region where the lipid was unperturbed [36]. Domain formation with lateral phase separation can increase motional disorder of the phospholipid acyl chains [37]. Such effects observed in MLVs of DMPC could be associated with the hemolytic effect reported for the peptide +2 (LC_50_ = 220 μM) [38]. Concomitantly, the transition enthalpy was reduced as the concentration of the peptide increased (Table 1), suggesting that less heat is needed for the transition from lβ’ to lα. This change of enthalpy in the main transition is due to an alteration at interactions between lipid acyl chains, as a result of the disruption of the intra- and intermolecular van der Waals interactions and trans-gauche isomerization, and this interpretation points to the insertion of the peptide, or at least a part of it, into the hydrophobic core of the membrane [18].

In a 1:25 peptide–lipid DMPC molar ratio, the peptide +5 exhibited a stronger effect on the phase transition than its analogue +2. In fact, the peak of the transition almost disappeared (Figure 2) and the transition enthalpy decreased from 1.47 to 0.18 J·g^−1^ (Table 1). Taking into account that these MLVs are made up of zwitterionic phospholipids, the net charge difference between both peptides would not be a variable to attribute such differences. The increase in the amphipathicity of the peptide +5 plays an important role in the strong effect observed on the transition, since this structural parameter can be related to an increased membrane affinity or an enhanced permeabilizing efficiency of the membrane-bound peptide fraction [39].

Regarding the MLVs (DMPC/DMPG 3:1) that mimic the surface characteristics of the bacterial membranes, these exhibited endothermic peaks for the pre-transition and main transition temperatures at 12 and 22.9 °C, respectively (Figure 3). After adding the peptides to these MLVs, the values of both Tm and the transition enthalpies decreased as the concentration of the peptide increased. At the maximum amount of the peptides (molar ratio peptide–lipid 1:25), the Tm decreased from 22.9 to 21.3 and 22.6 °C in the presence of the peptides +2 and +5, respectively, while the enthalpy was also reduced from 1.6 to 0.30 and 0.44, respectively (Table 1). The increased membrane fluidity and thermal destabilization of the transition could be associated with the antibacterial activity of both peptides against Gram-negative and Gram-positive bacteria reported in previous studies [12,13,38].

The effects of both peptides on the phase transition peaks of MLVs contained PC/PG mixture were similar to each other, suggesting that the role of the cationic charge of these peptides was not significant to disturb the membrane. However, the peptide with the reduced cationic charge (+2) induced a higher decrease of Tm than the peptide +5 (Table 1); hence, other structural variables may be related to the effect on membrane fluidity, such as hydrophobicity, which is an important parameter for the membrane of the peptide effect, as it controls the extent to which the peptide can partition into the hydrophobic core membrane [2]. Therefore, replacement of two alanines by two serines on the polar side of the peptide +2 reduced the ability to fluidize the MLVs of DMPC/DMPG.

### 3.3. Peptide +2 and +5 Secondary Structure before and after Interacting with Membrane Models

Figure 4 shows the ATR-FTIR absorbance (A) and second derivative (B) spectra of the peptide +2 and +5 in buffer solution at room temperature (RT). From the analysis of the amide I’ band (1700–1600 cm^−1^), it is possible to obtain information on the secondary structure of the peptides [24], while in the spectral region between 1600 and 1500 cm^−1^, the residual amide band II and bands due to the absorption of the lateral chains of amino acids can be identified. Analyzing the amide I’ region of the resolution-enhanced spectra, the bands associated to the different elements of the secondary structure of both peptides were identified and assigned. Peptide +2 shows a secondary structure composed mainly of β-elements that is apparent from the presence of characteristic bands associated to β-sheets arising at 1622 cm^−1^ (low-frequency β-sheets) and 1691 cm^−1^ (high frequency β-sheets), while the 1666 cm^−1^ peak can be assigned to turn elements [40]. Otherwise, in the peptide +5, the main peak of the amide I’ band is located at 1647 cm^−1^ and is due to the absorbance of unordered structures, while the two peaks that arise simultaneously at 1680 and 1618 cm^−1^ are a characteristic of the peptide aggregation due to the formation of intermolecular interactions [41,42]. Usually, peptides are unstructured and aggregate in aqueous solution but they may also form β-structures depending on both the nature of the peptide and the environment in which they are located [43]. The two peptides differ in 4 amino acids, which evidently give rather diverse properties. The presence of 2 further residues of arginine made the peptide +5 more positively charged than the peptide +2, in which there are 2 alanine residues that confer a higher hydrophobicity, while the presence of a residue of glutamic acid increases its negative charge. The different nature of the two peptides led the peptide +5 to show all features of a structureless and aggregated peptide in aqueous solution, while the peptide +2 assumed a mainly β-sheet conformation. In both peptides, the other bands arising below 1600 cm^−1^ were assigned to the residual amide II band (peak at 1540 cm^−1^) and to amino acid residues. In particular, the 1575 and 1527 cm^−1^ bands were due to histidine and lysine, respectively [44].

After studying the effect of the peptide on the thermotropic behavior of the membranes, we evaluated the effect of the membrane on the peptides secondary structure. ATR-FTIR spectra of both peptides were collected in the presence of multilamellar liposomes (MLVs) formed by POPC and POPC/POPG mixture. From the analysis and the comparison of the second derivative spectra of peptides +2 and +5, in the presence and absence of MLVs (Figure 5), it emerges that the binding to the membranes in both peptides induced structural changes. It is well known that membrane peptides have the characteristic to assume different conformations depending on the conditions and/or the type of membrane in which they are found. Generally, when these peptides are inserted into the membranes, they modify their conformation, forming α-helix structures that are energetically favorable in a hydrophobic environment such as the lipid bilayer [45]. In our case, in the presence of POPC model membranes, the peptide +5 showed a main peak at 1653 cm^−1^, which was due to α-helical structures absorbance [24], while the other peak that arose at 1684 cm^−1^, given the absence of a concomitant peak close to 1618 cm^−1^, can in this case be assigned to β-sheet structures, but it may also have contained information on turns since both turns and coupled high frequency vibrations of β-segments can contribute in the 1670–1690 cm^−1^ spectral region [46]. While the electrostatic lipid–peptide interactions appeared to induce an early partially folded state on the membrane surface, the hydrophobic interactions between the peptide nonpolar residues and the hydrophobic core of the lipid bilayer stabilized the α-helix upon membrane insertion [47], and in zwitterionic phospholipid membrane models, lacking the electrostatic contribution, these hydrophobic interactions can also lead to a different peptide folding [47,48]. In POPC/POPG MLVs, peptide +5 shows a secondary structure composed of β-sheet/turns and unordered structures as can be seen from the bands at 1681 and 1646 cm^−1^, respectively. An interesting property of some membrane peptides is their ability of helix-to-sheet conformation transition in membrane depending on their concentration and the lipid composition. In particular, zwitterionic phospholipids and low peptide concentrations promote that the peptide inserted into the membrane assumes a α-helix conformation [49,50], while the presence of negatively charged phospholipids and higher peptide concentrations can induce a conformational transition to β-sheets [49,51,52].

On the other hand, the same secondary structure elements can also be observed in the peptide +2 in the presence of both POPC and POPC/POPG membranes. Likely, the α-helix was inserted into the membrane, being conformation energetically favorable in a hydrophobic environment, while the β-conformation part of the peptide remained outside.

Going into more detail, the content of the secondary structure of the peptides alone and in the presence of liposomes was estimated by means of a curve-fitting procedure of the absorbance amide I’ bands (Figure 6). In this way, it was possible to perform a quantitative structural comparison between the two peptides and estimate the effect of the binding to the model membranes on their secondary structure. The curve-fitting confirmed that the secondary structure of the peptide +2 in aqueous solution and without the presence of any lipid showed a prevalence of β-structures content (66%) and a partial contribution of turns (34%), while the presence of both model membranes studied induced structural changes towards the formation of α-helices, which consisted of 64% in the case of POPC and 51% for POPC/POPG, and β-sheets/turns, which contributed with 36% and 49%, respectively. Unlike the peptide +2, peptide +5 alone showed all features of a structureless random coil, since about 50% of the area of the amide band I’ corresponded to unordered structures and the other 50% were due to the formation of intermolecular interactions. The presence of POPC model membranes induces a drastic structural change of the peptide that assumed a secondary structure composed of 72% α-helix and 28% β-elements and turns, while when POPG was added to the membrane, the main secondary structures were β-sheets and turns, being 81% of the total, with a lower contribution of 19% of unordered structures. A possible explanation for this behavior of peptide +5 in the presence of POPG is that the additional positive charges present in this peptide compared to peptide +2 allow the binding to the negatively charged surface of the membrane but prevent the peptide +5 penetration into the bilayer. This hypothesis will be addressed later with MD simulation. This would cause that the peptide is no longer unfolded and aggregated as in water solution and undergoes a conformational transition to a mainly β-structure due to the formation of electrostatic interactions with the membrane surface. The property of certain peptides to change their conformation from inserted α-helical form to surface-bound β-form, and vice versa, depending on the characteristics of the phospholipids that compose the membrane, was observed previously [49,52], and it was associated with the different functions that these peptides may have. All results of the curve-fitting are summarized in Table 2. Figure 6 also shows the component band due to the absorption of the C=O group of phospholipids, while in Table 2, the values of the ratio between the area of this band and the total area are reported. From these data, it is possible to roughly evaluate the affinity of each peptide for the two types of model membranes, since the C=O area/total area ratio is inversely proportional to the binding affinity. It is worth noting that peptide +5 has a greater affinity to both model membranes studied than peptide +2, which in turn, a negative charge being present in its amino acid sequence shows a lower ability to bind to the membrane of POPC/POPG. 

Absorbance spectra were used to calculate the fractional area component bands. χ^2^ values for the peptide +2 fittings are 8.2 × 10^−6^ in the absence of lipids, 8.0 × 10^−6^ in the presence of POPC, and 11.5 × 10^−6^ in the presence of POPC/POPG, whilst χ^2^ values for the peptide +5 fittings are 2.7 × 10^−6^ in the absence of lipids, 5.0 × 10^−6^ in the presence of POPC, and 9.1 × 10^−6^ in the presence of POPC/POPG. The values were rounded off to the nearest integer. The percentage value of the aggregation of the peptide +5 was obtained from the sum of the 1680 and 1618 cm^−1^ band areas, and the percentage value of the β-sheet component of the peptide +2 was obtained from the sum of the 1691 and 1622 cm^−1^ band areas. In the last column the ratios between the areas of the C=O group band of the phospholipids and the total areas are reported.

### 3.4. Molecular Dynamics Analysis of the Peptide-Membrane Complex

The first electrostatic interactions analysis comparing peptide +2 and peptide +5 was called “Affinity test”, and it was developed in order to know which peptide has the best attraction to the phospholipids surface in the vesicle membrane containing POPC/POPG mixture. The analysis was carried out using RMSD (root mean square deviation). At the beginning of MD simulation (120 ps), two separated molecular simulations were developed and are shown in a single figure, because the deviation located in POPG and POPC was very similar in both cases (Figure 7A). After starting the MD simulation and during the first 37 ps, the peptide +2 showed the best molecular interaction with the membrane. Furthermore, during that period of time the vesicle is not stable since 1560 molecules contained in vesicle need time to stabilize the electrostatic interactions in the established temperature. 

After 37 ps, POPC and POPG phospholipids exhibited more stability and the curve of the peptide +2 decreased, whereas that of the peptide +5 increased, suggesting an improved molecular attraction with the membrane. Special attention presents the zone from 90 to 112 ps, where the peptide +5 was close to the membrane surface (less than 1 Å) during 22 ps (Figure 7A). On the other hand, both peptides laid horizontally on the membrane surface and the carbon–carbon bonds were aligned at the same spatial position (Figure 7B). 

A longer molecular dynamic was performed to analyze the molecular insertion of peptide +5 into the membrane, and the results show that after 112 ps the membrane components had a change in the molecular deviation where POPC was separated from POPG (Figure 8A), suggesting that the peptide +5 induced a disorganization of the lipid chain order [54] in concordance with DSC results. Furthermore, the mechanism occurred when the side chains of the residues “LVKGIS” in peptide +5 sequence were drilling the volumetric Gaussian density map (Figure 8B).

Previous studies using molecular dynamics in a solvated model of peptide +5 without vesicle, found that the sequence GLKRIFKSGLGK was responsible for the highest electrostatic interactions in a smoothed electrostatic potential grid model [13], but under the vesicle influence and during the simulation, the sequence remained on the membrane providing stability, whereas the sequence LVKGIS was inserted towards phospholipid interfaces. On the other hand, AHVAS without α-helix structure did not interact with membranes. The incomplete partitioning of the peptide into lipid bilayer supports the previously established hypothesis in order to explain the β-sheet structure obtained by ATR-FTIR. Furthermore, the secondary structure trend of the peptide +5 interacting with PC/PG membranes obtained by MD and infrared spectroscopy was distinct (α-helix and β-sheet, respectively). Such structural differences could be because in the modelling studies, the peptide–lipid molar ratios used were 1:1560, whereas in the spectroscopic studies those molar ratios were 1:10. Previous studies have reported that low peptide concentrations promote that the peptide bound to the membrane assumes a α-helix conformation, while a higher peptide concentration can induce a conformational transition to β-sheets [49,52].

Specific molecular distances analysis were carried out using the best attractions between the residue isoleucine labeled by number 17 (located in the peptide +5) and a molecule of POPC interacting with peptide +5 (Figure 9A).

The analysis of the onset of the molecular interaction between the peptide +5 and choline molecule located in the polar head group of POPC finds that the distance variations are less than 1 Å (Figure 9A (red frame)). When oxygen from the phosphate group interacted with ILE 17, dipole–dipole forces were induced, and the big electronic cloud converts the side chain of ILE into a dipole transiently. After that, the LVKGIS sequence of the peptide +5 was pulled down into the membrane (Figure 9A (blue frame)), resulting from van der Waals forces. After 262 ps, the molecular interaction was not very clear (Figure 9A (green frame)) because the peptide +5 induced a disorganization of the lipid chain order [54], and the variation in the distances is increasing, but the peptide +5 remains on the membrane because the sequence GLKRIFKSGLGK provides the molecular stabilization outside the vesicle. Interestingly, in approximately 187 ps, the secondary structure (α-helix) of LVKGIS was destabilized to facilitate the membrane drilling (Figure 9B).

## 4. Conclusions

The substitutions of hydrophobic for hydrophilic/cationic residues at the polar face of Alyteserin 1c did not influence the thermotropic behavior of the membranes considerably. Thus, both peptides reduced the main phase transition temperature and the transition enthalpy, fluidizing the bilayer of the model membranes. Each peptide diluted in aqueous solution adopted a distinct secondary structure. β-sheet structures were predominant in peptide +2, while turns and unordered conformations were exhibited in peptide +5. However, after binding with model membranes, transitions toward α-helixes were observed except for the peptide +5 interacting with membranes containing PC/PG mixtures, which predominantly exhibited β-sheet structures. Finally, the MD simulations showed that the peptide–membrane interaction is a dynamic system depending on the time. The MD results supported that the sequence LVKGIS of the peptide +5 had the ability to insert contained PC/PG lipids into the surface of the membrane, specifically by interaction between ILE17 and POPC. Meanwhile, another part of the sequence “GLKRIFKSGLGK” remained on the membrane providing stability. Both behaviors led to the fluidization of the bilayer.

## Figures and Tables

**Figure 1 biomolecules-09-00527-f001:**
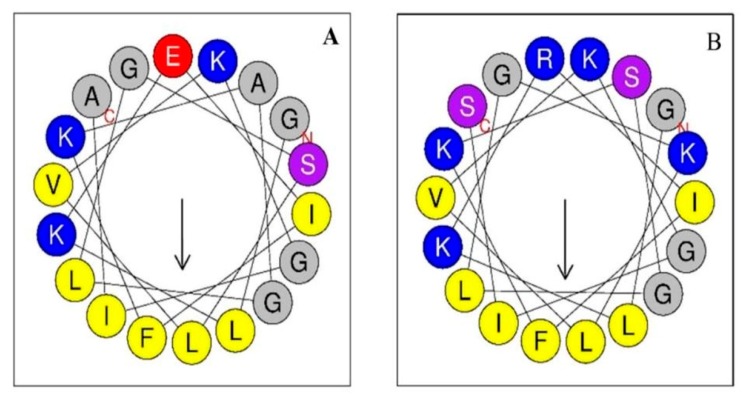
Wheel projections of the first 18 residues of the sequence of each peptide. Alyteserin-1c peptide (+2) (**A**); peptide +5 (**B**). The hydrophobic amino acids are yellow, and the charged amino acids are blue (positive charge) or red (negative charge). The polar amino acids are purple and those in between are grey.

**Figure 2 biomolecules-09-00527-f002:**
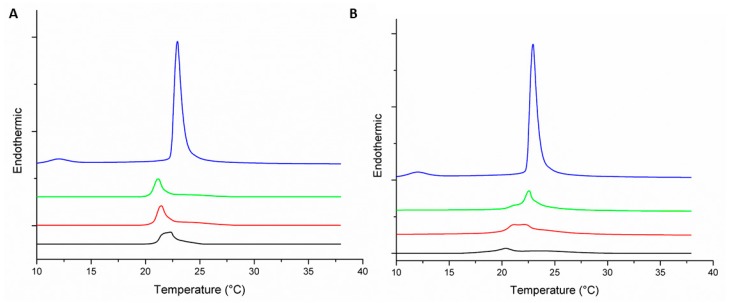
Differential scanning calorimetry (DSC) thermograms representing endothermic peaks of the phase transition of multilamellar vesicles (MLVs) formed by 1,2dimyristoyl-sn-glycero-3-phosphocholine (DMPC) under the effect of peptide +2 (**A**) and peptide +5 (**B**) at different peptide–lipid molar ratios: 0:1 (

); 1:100 (

); 1:50 (

); and 1:25 (

).

**Figure 3 biomolecules-09-00527-f003:**
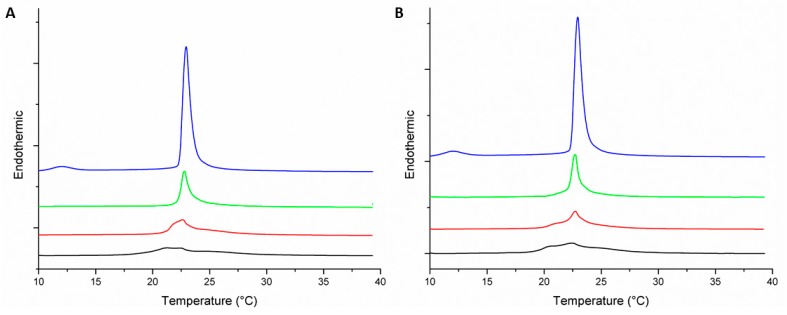
Thermotropic behavior of DMPC/DMPG (3:1) MLVs under different amounts of peptide +2 (**A**) and peptide +5 (**B**). Peptide amounts were tested at different peptide–lipid molar ratios: 0:1 (

); 1:100 (

); 1:50 (

); and 1:25 (

).

**Figure 4 biomolecules-09-00527-f004:**
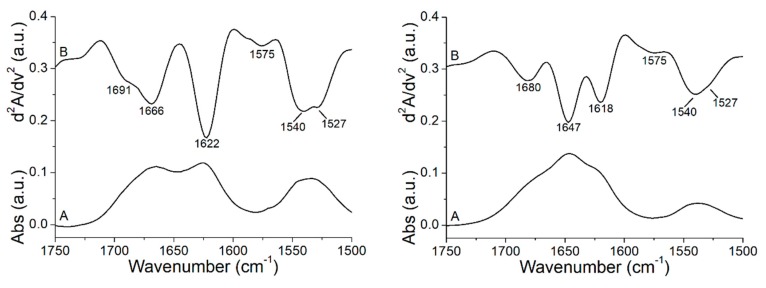
Original absorbance (A) and second derivative (B), FTIR spectra of the peptide +2 *(***left***)* and +5 *(***right***)* between 1750 and 1500 cm^−1^ at RT.

**Figure 5 biomolecules-09-00527-f005:**
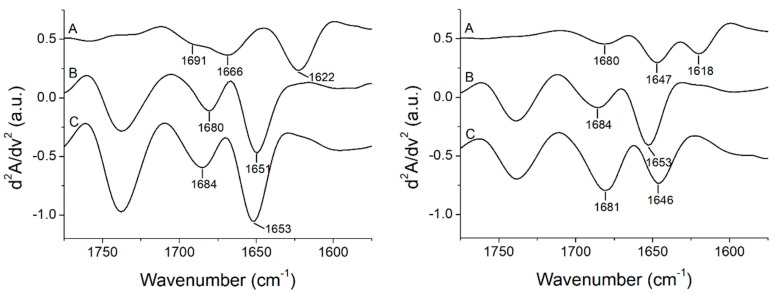
Stacked second derivative spectra of the peptide +2 *(***left***)* and +5 (**right**) in the absence of lipids (A) and in the presence of multilamellar vesicles of POPC (B) and POPC/POPG (C). The spectra are reported in the 1775–1575 cm^−1^ region to show the spectral range of both the lipids’ carbonyl group and the peptides’ amide I’ bands.

**Figure 6 biomolecules-09-00527-f006:**
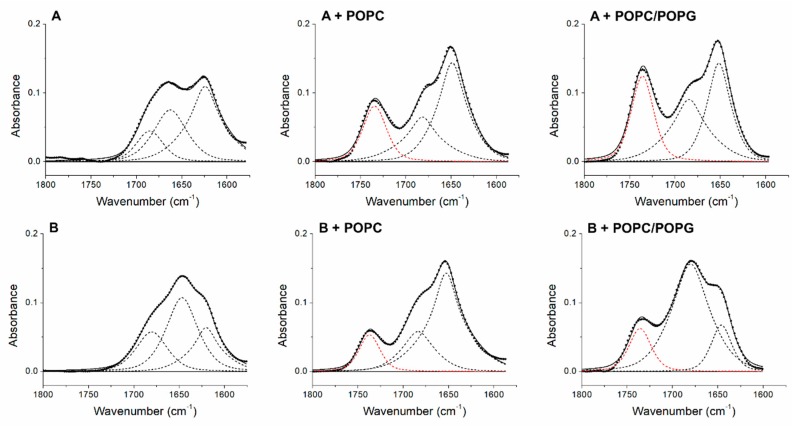
Amide I′ band decomposition of the peptide +2 (**A**) and +5 (**B**) alone and in the presence of MLVs composed of POPC and POPC/POPG. Dashed lines are the individual Voigt bands fitted to the spectrum. The dotted lines represent the original absorbance amide I′ bands and the continuous lines are the sum of the individual components. The red dashed lines are the C=O group absorption bands of the phospholipids. The numerical values are reported in Table 2.

**Figure 7 biomolecules-09-00527-f007:**
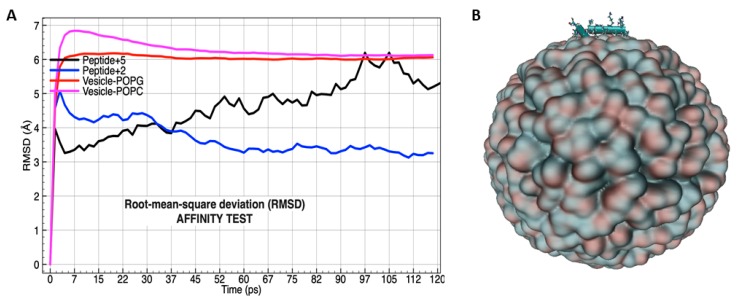
RMSD (root mean square deviation) by affinity test between peptide +2 (blue line) and peptide +5 (black line) in accordance with POPG and POPC (red and magenta lines, respectively) phospholipids in vesicle conformation (**A**). Molecular orientation of peptide onto lipid bilayer conformation in double layered spherical vesicle POPC and POPG (**B**). Peptide is shown by CPK coloring convention, and the vesicle by the drawing method that use the isosurface extracted from a volumetric Gaussian density map, called: QuickSurf [53].

**Figure 8 biomolecules-09-00527-f008:**
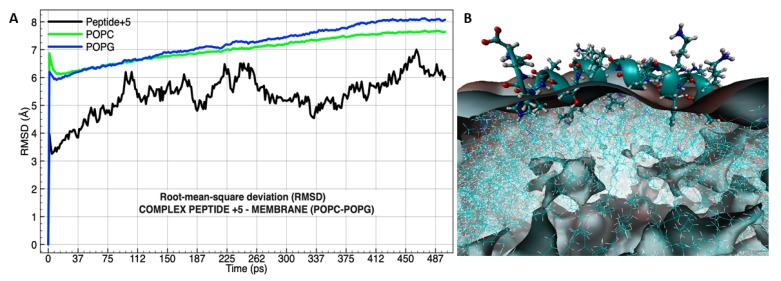
RMSD (root mean square deviation) in 500 ps of molecular dynamic simulation of the peptide +5 (black) with phospholipids POPG (blue) and POPC (green) (**A**); graphical representation of the peptide +5 insertion into the membrane ((**B**), image taken at 97 ps). Peptide and phospholipids are shown by the graphical representation using CPK coloring convention, the first represented by bonds and the second by lines. The model allows to see the α-helix structure and everything influenced by QuickSurf [53].

**Figure 9 biomolecules-09-00527-f009:**
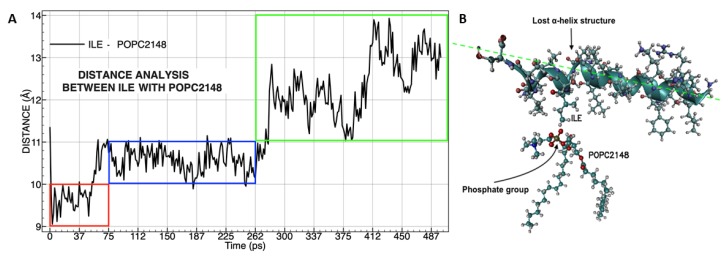
Molecular distances analysis between isoleucine (ILE17) with phosphatidylcholine (**A**), the color frames represent three times of affinity during molecular dynamics. Molecular representation was taken during peptide insertion into the membrane (**B**). The specific forces of van der Waals called dipole–dipole forces are represented between the phosphate group (polar) in POPC and the side chain (non-polar) of ILE (C-H----O-P). The green dotted line shows the change of the spatial position of peptide +5 that causes the loss of α-helix structure and displacement of the LVKGIS sequence.

**Table 1 biomolecules-09-00527-t001:** Changes in the values of the Tm and ΔH, as result of the interaction of the peptides +2 and +5 at different amounts with MLVs constituted by DMPC and DMPC/1,2-dimyristoyl-sn-glycero-3-phosphoglycerol (DMPG) (3:1).

MLV	Peptide–Lipid Molar Ratio	Pre-Transition Temperature (°C)	Tm (°C)	ΔH (J·g^−1^)
DMPC	0:1	12.9	22.9	1.47
DMPC-Peptide +2	1:100		21.2	0.63
	1:50		21.5	0.62
	1:25		21.2	0.62
DMPC-Peptide +5	1:100		22.7	0.8
	1:50		22.2	0.63
	1:25		20.5	0.18
DMPC/DMPG (3:1)	0:1	12	22.9	1.6
DMPC/DMPG (3:1)-Peptide +2	1:100		22.9	0.54
	1:50		22.8	0.38
	1:25		21.3	0.30
DMPC/DMPG (3:1)-Peptide +5	1:100		22.9	0.64
	1:50		23	0.45
	1:25		22.6	0.44

**Table 2 biomolecules-09-00527-t002:** Secondary structure composition of the peptide +2 and +5 in the absence and in the presence of model membranes as calculated by curve fitting of the amide I′ band at RT.

Peptide	α-Helix (%)	β-Sheet (%)	Turns (%)	Unordered (%)	Aggregation (%)	C=O Band Area/Total Area (%)
+2		66	34			
+2 + POPC	64	36				22
+2 + POPC/POPG	51	49				29
+5				48	52	
+5 + POPC	72	28				14
+5 + POPC/POPG		81		19		16

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
