# Peer review of "Studies on the Interaction of Alyteserin 1c Peptide and Its Cationic Analogue with Model Membranes Imitating Mammalian and Bacterial Membranes"

_biomolecules, 2019, doi:10.3390/biom9100527_

Round 1

Reviewer 1 Report

Dear Authors,

The manuscript titled, "Studies on the interaction of Alyteserin 1c peptide
 and its cationic analogue with model membranes imitating mammalian and bacterial membranes" is a well designed study and interesting piece of research. It is interesting to see the conformational switch of the peptide in different lipid environments. 

I recommend for publication without further review.

Author Response

Thank you very much for your comments!!

Reviewer 2 Report

Review of the manuscript biomolecules-593019-peer-review-v1

Title: Studies on the interaction of Alyteserin 1c peptide and its cationic analogue with model membranes imitating mammalian and bacterial membranes.

The antimicrobial peptide (AMP) Alysterin and his analogue were studied by different biophysical methods. The paper has a logical structure and it is clearly written. The changes of thermodynamic behavior of the model membrane due to interaction are reported for both studied peptides. Also their conformation changes were studied using infrared spectroscopy and the results are complemented by the MD simulations.

Nevertheless I have some comments, questions and suggestions, namely as regard to part dealing with the infrared spectroscopy. The authors in the chap. 2.2 described the standard peptides synthesis, where for purification TFA in the mobile phase is used. As TFA has strong infrared absorption at amid I spectral region (1673cm-1) to get clear IR spectra of peptides/proteins in amid I spectral region there are two option – either to subtract this spectral contribution or to perform exchange of contra ions TFA/Cl as describe for example in. J Pept Sci. 2007 Jan;13(1):37-43. None of these options are mentioned in the text and obviously all interpretation could be influenced by presence of TFA ions. As regard IR experiment the authors disused amid I’ spectral band, which represents carbonyl vibration of peptide bond only when D2O solution is used, but no such possibility is mentioned in experimental section. On the other hand if the H2O solution is used for such experiments, for baseline subtraction the water band at around 2126 cm-1is used (H. Fabian, W. Mantele - Infrared Spectroscopy of Proteins in Handbook of Vibrational Spectroscopy 2006 John Wiley & Sons, Ltd), thus I am not sure if already baseline substraction as described in Chap. 2.6 was correctly performed. Anyway I would recommend the publication of this work with major revision with the respect to comments namely as regard the infrared experiments, where all experimental conditions should be clarified and if necessary the results revisited.

Author Response

Thank you for yours valuable comments.

"As TFA has strong infrared absorption at amid I spectral region (1673cm-1) to get clear IR spectra of peptides/proteins in amid I spectral region there are two option – either to subtract this spectral contribution or to perform exchange of contra ions TFA/Cl"

Answer: The peptides, after being purified using TFA as the mobile phase, were dried and subsequently lyophilized (adhered in the manuscript). As can be confirmed in the attached mass spectra of the peptides, there are no peaks corresponding to TFA ions, or the appearance of adducts that can alter the molecular mass of the peptides, suggesting a total removal or a negligible remanence of the solvent, which may interfere with the analysis of the amide I band.

"As regard IR experiment the authors disused amid I’ spectral band, which represents carbonyl vibration of peptide bond only when D2O solution is used, but no such possibility is mentioned in experimental section."

Answer: Your statement is really true and we appreciate for your suggestions. The liposomes were hydrated with D2O and it was cleared in the manuscript. As can be confirmed in the attached spectra of POPC/peptide +5, is present the band at 2500 cm-1 (typical of deuterated water) and that of normal water that would be around 3300 cm-1 is absent.

" if the H2O solution is used for such experiments, for baseline subtraction the water band at around 2126 cm-1is used (H. Fabian, W. Mantele - Infrared Spectroscopy of Proteins in Handbook of Vibrational Spectroscopy 2006 John Wiley & Sons, Ltd), thus I am not sure if already baseline substraction as described in Chap. 2.6 was correctly performed"

Answer: normal water was not used. 

Reviewer 3 Report

Authors describe an AMP isolated from Alytes obstetricans (+2) and its more cationic and hydrophilic analogue (+5). They study the interaction with model membranes by calorimetric and spectroscopic assays. They observed different transitions from these peptides after binding with model membranes. The study is sound and well designed. Furthermore, they performed molecular dynamics (MD) simulations to study this difference.

Author Response

Thank you very much for your comments!!

Round 2

Reviewer 2 Report

I have no additional comments.